# Improvement in the Chromium(VI)-Diphenylcarbazide Determination Using Cloud Point Microextraction; Speciation of Chromium at Low Levels in Water Samples

**DOI:** 10.3390/molecules29010153

**Published:** 2023-12-26

**Authors:** Begoña A. Mouco-Novegil, Manuel Hernández-Córdoba, Ignacio López-García

**Affiliations:** Department of Analytical Chemistry, Faculty of Chemistry, Regional Campus of International Excellence “Campus Mare-Nostrum”, University of Murcia, E-30100 Murcia, Spain; bamouco@gmail.com (B.A.M.-N.); hcordoba@um.es (M.H.-C.)

**Keywords:** chromium, chromium speciation, diphenylcarbazide, cloud point extraction, molecular absorption spectrometry, water samples

## Abstract

A reliable, rapid, and low-cost procedure for determining very low concentrations of hexavalent chromium (Cr) in water is discussed. The procedure is based in the classical reaction of Cr^6+^ with diphenylcarbazide. Once this reaction has taken place, sodium dodecylsulfate is added to obtain an ion-pair, and Triton X-114 is incorporated. Next, the heating of the mixture allows two phases that can be separated by centrifugation to be obtained in a cloud point microextraction (CPE) process. The coacervate contains all the Cr^6+^ originally present in the water sample, so that the measurement by molecular absorption spectrophotometry allows the concentration of the metal to be calculated. No harmful organic solvents are required. The discrimination of hexavalent and trivalent forms is achieved by including an oxidation stage with Ce^4+^. To take full advantage of the pre-concentration effect inherent to the coacervation process, as well as to minimize reagent consumption and waste generation, a portable mini-spectrophotometer which is compatible with microvolumes of liquid samples is used. The preconcentration factor is 415 and a chromium concentration as low as 0.02 µg L^−1^ can be detected. The procedure shows a good reproducibility (relative standard deviation close to 3%).

## 1. Introduction

The determination of Cr^6+^ in water is a matter of interest, as the concentrations usually present are very low, and legislation is becoming more and more demanding via decreasing the maximum permitted concentration of this species of recognized toxicity [1,2]. The classical procedure based on the spectrophotometric measurement of the reaction product with 1,5 diphenylcarbazide (DPC) has an excellent selectivity but does not reach the required sensitivity [3,4,5,6]. Therefore, a variety of procedures using atomic absorption spectrometry or plasma-based techniques have been proposed for this purpose [7,8,9,10]. Most of the more recent ones have in turn been combined with microextraction procedures that concentrate the analyte in a low volume of liquid and thus further increase sensitivity [11,12,13,14,15]. Although the reliability of these atomic or mass techniques is beyond doubt, they require the use of gases and instrumentation of some complexity in the laboratory. From a practical point of view, it would be interesting to have a procedure involving simple, low-cost instrumentation available in all laboratories, such as a modest UV–visible molecular absorption spectrophotometer.

As indicated, the classical reaction with DPC does not present sufficient sensitivity [16] but current advances in microextraction techniques can considerably improve this analytical parameter. In this respect, cloud point extraction is particularly attractive as very high pre-concentration factors of the analyte can be achieved [17,18,19]. This methodology is based on the formation of condensed phases (coacervates) from surfactant-containing solutions. A simple heating step, or variation in the salt content, produces small volumes of coacervates which are separated by centrifugation. These surfactant-rich phases have physicochemical properties that are very different from those of the aqueous solutions in which they are formed and can extract low polarity compounds without the use of organic solvents. This provides a basis for the determination of metals through the extraction of one of their neutral complexes, followed by an appropriate analytical measurement of the coacervate [20,21,22].

Coacervation is the phenomenon whereby a homogeneous colloidal solution separates into two immiscible liquid phases. One of the phases is rich in colloids and is called the coacervate. Surfactants are often used as colloids to induce coacervation. In this case, the appearance of inter-colloidal interactions that form micelles is promoted by, among other things, changing the temperature of the solution. The result is the formation of a cloudy suspension of amorphous droplets that tend to coalesce into a liquid phase that is either on top or on the bottom, depending on its density [23]. The surfactant must be added at a concentration above its critical micellar concentration (CMC) [24] and the solution temperature must be raised above the cloud point temperature (CPT) [25] of the surfactant. The result is the formation of two isotropic phases with very different amounts of surfactant. Phase separation occurs because the polar groups of the surfactant dehydrate, reducing the repulsion between the micelles and increasing their interaction when the CPT is reached. This temperature depends on the chemical structure of the surfactant. It can be as low as 23 °C for Triton X-114 or as high as 64 °C for Triton X-100. When surfactants are dissolved in water at appropriate concentrations and temperatures, amphiphilic molecules self-assemble into structures that keep their hydrophilic head groups on the outside and hydrophobic tails on the inside away from water. The hydrophobic core of the micelle can then bind to hydrophobic compounds as an ion pair, separating it from the liquid phase.

In recent years, this methodology has been used for the extraction of various Cr^6+^ complexes, measuring the final concentration of the metal in the condensed phase by means of an atomic [26,27,28,29,30,31] or molecular techniques [11,16,26,32,33,34].

An analytical procedure based on this general approach is presented here. Cr^6+^ reacts with diphenylcarbazide to form a complex of Cr^3+^ with diphenylcarbazone (DPCO) [11]. The compound resulting from the reaction is positively charged [35] and, in the presence of a bulky anion such as that of sodium dodecylsulfate (SDS), in fact gives rise to a neutral ionic pair which is extracted in the small volume coacervate originating from a neutral surfactant (Triton X-114). The measurement of the molecular absorbance of this colored species in the condensed phase allows the determination of Cr ^6+^ with excellent sensitivity. This idea has been explored by others using chloride as the counterion to form the ion pair and Triton X-100 as the surfactant for the CPE. In that case, the presence of NaCl is required to achieve an ionic strength that allows the formation of the coacervate, but an excess interferes with its stability [26]. To the best of our knowledge, the idea has not been explored from this perspective and only one procedure has been reported in which SDS is used but no CPE is applied, and where instead it is extracted by means of a more conventional dispersive microextraction with the aid of an organic solvent [36].

## 2. Results and Discussion

This process is based on the oxidation of DPC to diphenylcarbazone, induced by the presence of Cr^6+^ in an acidic medium. The very reactive Cr^3+^ resulting from the reduction in the hexavalent species of chromium is capable of forming a colored complex with the oxidation product 1,5-diphenylcarbazone (λ_max_ = 540 nm) [35,37,38]. This method is highly selective for Cr^6+^ in the presence of other metals [5,6,34].

First, the experimental conditions which are recommended in the literature [35,39] for the direct determination of Cr^6+^ with DPC were revised, checking the stability of the color and maximum sensitivity at a relatively high analyte concentration (1 µg mL^−1^).

As is known [35], this reaction takes place in two stages: first, DPC is oxidized to 1,5-diphenylcarbazone (DPCO) by Cr^6+^, and then the Cr^3+^-DPCO complex is formed. The first step takes place at pH = 5, but the colored compound is unstable because it evolves into ${Cr(OH)}_{2}^{-}$. The study of the effect of the pH of the solution on the intensity of the colour formed with DPC is shown in Figure 1A. As can be seen, the best results are obtained by using a pH close to 1. In our case, we obtained excellent results by adding 100 µL of concentrated nitric acid to 10 mL of sample.

As shown in Figure 1B, under these acidic conditions, the concentration of DPC required to obtain maximum signals must be at least 2.5 × 10^−4^ mol L^−1^. This concentration is achieved by adding 50 µL of 0.05 mol L^−1^ DPC to 10 mL of solution. In these conditions, the characteristic purple color of the complex was at a maximum after a 15-min period. The graph inset of Figure 2 shows the kinetics of the formation of the colored compound. These were the starting conditions used to study the microextraction process at very low chromium concentrations. Some species such as bromide [40] and trace metal ions such as Cu^2+^ [41] are known to catalyze this reaction; however, the experimental conditions to assess their effect are not suitable for the determination of chromium.

### 2.1. Optimization of the CPE Stage

Since the Cr^3+^-DPCO complex is positively charged, a negative large-size counter-ion is needed to give rise to a neutral ion-pair in the solution. With this in mind, a large excess of SDS was added and the mixture heated until above the cloud point temperature of the surfactant, but no suitable CPE process was observed. When the ionic strength was increased through the addition of sodium chloride, the formation of a condensed phase was achieved by heating above the cloud point of SDS. However, ion pair extraction did not occur. The experiments were repeated including an excess of cationic surfactants. Namely, hexadecylpyridinium chloride, hexadecyltrimethylammonium bromide, and tetradecyltrimethylammonium bromide were assayed but all the attempts to extract the colored complex in a condensed phase failed.

On the contrary, when, in addition to the large excess of SDS, neutral surfactants were incorporated into the mixtures and then heated, condensed phases were obtained. Four non-ionic surfactants were assayed (IGEPAL CA-630, Triton X-45, X-100, and X-114) and Triton X-114 proved the most suitable for the purpose since the condensed phase obtained showed an intense color with maximum absorption at 546 nm, as a consequence of the extraction of the Cr^3+^-DPCO complex. SDS acts here as the counter-part of the ion-pair in the solution, while the surfactant-rich phase originated by Triton X-114 plays the role of the organic solvent in a more conventional dispersive microextraction process.

Figure 2 shows spectra obtained from a 2 µL nanodrop placed in the stand of the spectrophotometer, while Figure 3 shows the results obtained for the optimization of the concentrations to be used for both surfactants. The results proved that in the optimized experimental conditions, i.e., 0.04% SDS and 0.2% Triton X-114, the analyte was quantitatively transferred to the coacervate. High concentrations of SDS caused a decrease in signal due to the increase in volume in the micellar phase.

### 2.2. Analytical Figures of Merit

Using the optimized procedure provided in the Materials and Methods section, a linear relationship between the analytical signal obtained from the coacervate by means of the nanodrop spectrophotometer and the chromium concentration was verified in the 0.04–2 µg L^−1^ chromium range. To calculate the repeatability, three Cr^6+^ solutions prepared at concentrations within the linear range, namely 0.2, 0.5, and 1 µg L^−1^ were submitted to the procedure, each coacervate being measured ten times. The relative standard deviations were 3.0, 2.8, and 2.5%, respectively. The detection limit calculated using the criterion based on three times the standard error of the regression line was calculated to be 0.02 µg L^−1^ with a determination limit of 0.04 µg L^−1^ chromium. It is important to note that these values were obtained when 10 mL sample aliquots were used. The sensitivity can be further increased via an increase in the sample volume. We verified this point by using 20-mL aliquots. Larger volumes could be used but in this case, the Triton X-114 amount should be revised since the final volume recovered after coacervation would vary. The 10 mL sample volume is recommended since the sensitivity is sufficient to verify a suitable control of the analyte at the levels indicated in the present legislation. The enrichment factor was calculated from the ratio of the slopes of calibration lines obtained by measuring the absorbance in the coacervate with those found directly from aqueous solutions that were not submitted to the CPE stage, and it was found to be 415.

On the other hand, to provide an alternative procedure for those laboratories in which a nanodrop instrument is not available, experiments were also carried out to adapt the procedure for measurements in a common spectrophotometer. In our opinion, the most suitable way for this purpose is to use a flow cell and to fill the inner volume with the coacervate with the aid of a syringe. To facilitate this handling of the high-viscosity condensed phase, 10 µL was diluted with 10 µL of acetone. This means that the enrichment factor decreases, and the detection limit increases, but the sensitivity is still high and gives a response to the analytical problem. Even more, taking advantage of this high sensitivity, in the case of unavailability of a flow cell, a common low volume 1-cm optical path cell could be used for diluting properly the coacervate.

Table 1 summarizes the main analytical characteristics of the procedure studied here together with those of others reported in the literature for the same purpose. It should be noted that this approach using Cr^6+^-DPC-CPE with color measurement has a limit of determination comparable to others based on atomic techniques, involves lower cost instrumentation, and is more environmentally friendly as it saves on reagents and generates very little waste, all without the need to use organic solvents or gases.

### 2.3. Speciation Results

Once the conditions for Cr^6+^ determination had been optimized, the potential usefulness of the procedure for chromium speciation was considered. This topic, which has been addressed by several authors through the oxidation of Cr^3+^ [15,20,42,43], was re-examined here as the analyte concentrations are very low and there is a possibility that excess oxidant could react with the DPC, thus altering the speciation result.

Firstly, the action of three oxidants, namely potassium peroxydisulfate [33,43], concentrated hydrogen peroxide [44,45,46], diluted potassium permanganate [35,44], and diluted Ce^4+^ nitrate [3,47] solutions, was tested separately on relatively concentrated (1 µg L^−1^) Cr^3+^ solutions [43,47,48] in the presence and absence of DPC. The first of the oxidants mentioned was discarded, as it proved unsuitable for the purpose. Permanganate solution, which has been suggested by others [42], can achieve oxidation, and its excess can be removed by the addition of sodium azide. However, the oxidation treatment is not convenient as prolonged heating is required for the reaction to be complete. Excellent results were obtained when the oxidation was carried out in a nitric acid medium with a dilute Ce^4+^ solution (0.05 mol L^−1^), the excess of which did not affect either the DPC or the other reagents.
molecules-29-00153-t001_Table 1Table 1Comparison of the proposed procedure with others published in the literature using diphenylcarbazide and several microextraction and/or determination techniques.AnalyteSeparationDetectionCalibration, µg L^−1^LOD, µg/LEFReferenceCr^3+^, Cr^6+^SEETAAS1–100.2--[6]Cr^3+^, Cr^6+^SM-DLLME-SFOUV–Vis1–400.2350[11]Cr^3+^, Cr^6+^NoneUV–Vis500–30,000; 5–300300; 3--[16]Cr^6+^CPEUV–Vis; FAAS1.5–250; 1.85–10000.41; 0.555; 10[26]Cr^3+^, Cr^6+^FIA-CERUV–Vis0–270–120 pg--[33]Cr^6+^SPEUV–Vis0–260.1550[34]Cr^3+^, Cr^6+^EMEETAAS0.05–50.02110[35]Cr^3+^, Cr^6+^IP-SA-DLLMEFO-LADS0.2–200.05159[36]Cr^6+^SPMEUV–Vis1.8–600.6125[49]Cr^3+^, Cr^6+^UEAASLLM-SFOETAAS0.01–0.30.003174[50]Cr^3+^, Cr^6+^SA-DLLME–SFOETAAS0.05–0.40.00440[51]Cr^3+^, Cr^6+^CPEUV–Vis0.04–20.02415This workEF: enrichment factor; SE: solvent extraction; ETAAS: electrothermal atomic absorption spectrometry; SM-DLLME-SFO: supramolecular dispersive liquid–liquid microextraction method based on the solidification of floating organic drops; FAAS: flame atomic absorption spectrometry; FIA-CER: flow injection analysis–cation exchange resin packet; SPE: solid phase extraction; EME: electromembrane extraction; IP-SA-DLLME: ion pair-based–surfactant-assisted dispersive liquid–liquid microextraction; SPME: solid phase microextraction; UEAASLLM-SFO: ultrasound-enhanced air-assisted surfactant liquid–liquid microextraction based on the solidification of a floating organic droplet; SA-DLLME–SFO: surfactant-assisted dispersive liquid–liquid microextraction based on the solidification of floating organic drop; FO-LADS: fiber optic–linear array detection spectrophotometry.


Figure 4A shows the results obtained by studying the reaction time of Ce^4+^ with Cr^3+^. The kinetics were followed by the reaction of the Cr^6+^ formed with DPC. In all cases, the work was carried out at room temperature. As can be seen, after 15 min of reaction, the application of the proposed procedure for the determination of Cr^6+^ generated by the oxidation process leads to maximum and constant signals. Therefore, this time is recommended as sufficient to carry out the oxidation.

Regarding the acidic medium, the use of both nitric [3] and sulfuric [43] acid is recommended to carry out the oxidation. Similar experiments have been carried out with similar concentrations of both acids. After 15 min of reaction, slight differences were observed between the solutions prepared with both acids, the reaction yield being higher in the case of using nitric acid. Therefore, HNO_3_ was selected as the optimal acid medium for both Cr speciation and complexation with DPC.

Figure 4B shows the results obtained by varying the concentration of Ce^4+^ used for the oxidation of Cr^3+^ on the absorbance signal measured in the micellar phase using the proposed procedure. As can be seen, the presence of a high concentration of Ce^4+^ causes a loss of sensitivity. However, if the concentration of Ce^4+^ does not exceed the value of 3 × 10^−4^ M, the signal is not affected by the presence of this oxidizing agent, making it unnecessary to use another chemical for the elimination of the remaining excess Ce^4+^. Therefore, the use of Ce^4+^ 2 × 10^−4^ M as the oxidizing agent under the proposed conditions is recommended. Higher concentrations of Ce^4+^ cause a slight decrease in the signal. This is due to a slight interference in the reaction of Cr^6+^ with DPC.

The colored product obtained showed a molar absorptivity of 4.447 × 10^4^ mol^−1^ L cm^−1^ agreeing with that obtained experimentally for a Cr^6+^ solution of the same concentration under the same experimental conditions, which was calculated as 4.436 × 10^4^ mol^−1^ L cm^−1^, thus demonstrating the transformation was complete.

The reliability of the speciation at very low concentrations was tested experimentally by preparing a series of solutions of varying Cr^6+^ and Cr^3+^ concentrations which were subjected to the complete speciation process. In each of these, oxidation was first carried out with an aliquot to determine the total chromium content, and the Cr^6+^ content was determined in a second aliquot. The results of these experiments, carried out in triplicate, are summarized in Table 2.

The AGREEprep tool [52] was used to assess the greenness of the proposed procedure. The study of the ten criteria for the evaluation of the sustainability shows that the proposed procedure resulted in an overall score of 0.67. This value is comparable to other microextraction methods. A procedure is considered green if its total score is greater than 0.6 [53].

In addition, as a metric tool to assess the practicality of an analytical method, the Blue Applicability Grade Index (BAGI) [54] was used. In this tool, the overall result of the evaluation is a pictogram with a number in the center that indicates the overall score assigned to the analytical method. This value ranges from 25 to 100. A method is considered practical if it scores at least 60 points. The hue scale of the pictogram shows the conformity of the method with the established criteria. The application of this tool to the proposed procedure resulted in an overall score of 72.5, indicating that the overall protocol has good potential applicability. Figure 5 shows both pictograms.

### 2.4. Application of the Proposed Procedure to Water Samples

The proposed procedure for the determination of Cr^6+^ and Cr^3+^ has been applied to water samples of different nature. In all cases, a recovery test was applied to both species. As can be seen in Table 3, the recovery results are excellent.

The validity of the procedure was evaluated by analyzing the total chromium contents and those of Cr^3+^ and Cr^6+^ species in four reference materials. When the total content was higher than the linear calibration range, dilution was carried out accordingly. The results obtained are shown in Table 4. As can be seen, the content found for total chromium does not show significant differences with respect to the certificate.

## 3. Materials and Methods

### 3.1. Instruments

A Thermo Scientific NanoDrop™ model 1000 spectrophotometer (Willmington, DE, USA), suitable for working between 220 and 750 nm, was used together with version 3.8 of its software. For comparative purposes, an ATI UNICAM spectrophotometer model UV2 (Cambridge, UK) equipped with a 50 µL flow cell from Hellma (Müllheim, Germany) was occasionally used. A thermostatic bath with 50 W ultrasonic power (ATU, Valencia, Spain) was used for sample heating. Centrifugation was performed in a Unicen 21 (Madrid, Spain) centrifuge equipped with a conical tube adapter capable of working at 4000 rpm (1540 g). Pipetting of small volumes of solution was carried out with single-channel electronic micropipettes of different variable volumes provided by Thermo Scientific.

### 3.2. Reagents

Pure water (18 MΩ·cm resistivity) obtained from a Milli-Q system (Millipore, Belford, MA, USA) was used in the preparation of the solutions. Plastic and glassware were washed with diluted nitric acid (1% *v*/*v* concentrated acid) before use. Standard Cr^3+^ and Cr^6+^ (1 g L^−1^) solutions were prepared from solid Cr(NO_3_)_3_·9H_2_O and K_2_Cr_2_O_7_ (Fluka, Buchs, SG, Switzerland), respectively, and diluted conveniently daily to prepare working solutions. The reagent 1,5-diphenylcarbazide, DPC, (0.05 mol L^−1^) was prepared by weighing the required amount of product (Sigma-Aldrich, Steinheim, Germany) and dissolving it in acetone. The surfactants sodium dodecylsulfate (SDS), hexadecylpyridinium chloride, hexadecyltrimethylammonium bromide, tetradecyltrimethylammonium bromide, IGEPAL^®^, and Triton X-45, X-100, and X-114 and the rest of the reagents used were obtained from Sigma-Aldrich.

### 3.3. Samples and Reference Materials

Four water samples were analyzed. First, a sample of tap water taken in the laboratory. One sample of water was taken from a natural spring and two samples were comprised of bottled water that was purchased at a local supermarket. All samples were filtered through a 50 µm sieve and kept at 4 °C in plastic bottles.

In addition, four reference materials were used, namely a spring sample (SRM 1640a), two water samples from Lake Ontario with different levels of trace element fortification (TM-25.4 and TM-23.4), and a rainwater sample from Ontario (TMRain-04). These reference materials were obtained from the National Institute of Standards and Technology and the Research Council of Environment Canada.

### 3.4. Procedures

In a 15 mL centrifuge tube, 10 mL of Cr^6+^ solution containing 0 to 2 µg L^−1^ Cr^6+^ was placed and 100 µL of concentrated nitric acid and 50 µL of 0.05 mol L^−1^ DPC solution was added. The mixture was left to react for 15 min. Then, 400 µL of 1% SDS and 100 µL of 20% Triton X-114 were added. The solution was homogenized and heated to 60 °C for 10 min. The condensed phase was recovered at the bottom of the tube after centrifugation for 10 min at 3500 rpm. The upper aqueous phase was discarded, and 2 µL of condensed phase was taken and placed in the spectrophotometer stand so that the signal at 546 nm could be measured. Calibration was carried out through applying the same procedure to Cr^6+^ standard solutions.

For the speciation purpose, 10 mL of sample was taken, to which 120 µL of concentrated nitric acid solution and 25 µL of 0.05 mol L^−1^ Ce^4+^ solution were added. The oxidation of Cr^3+^ to Cr^6+^ was allowed to proceed for 10 min. Next, 50 µL of 0.05 mol L^−1^ DPC solution was added and the Cr^6+^ determination was continued as above. The analytical signal finally obtained in the spectrophotometer corresponded to the total chromium content. The Cr^3+^ content was determined by difference. 

## 4. Conclusions

The difficult challenge of the determination of very low concentrations of Cr^6+^, as well as the speciation of this element, can be solved with the use of low-cost instrumentation, such as a simple visible–UV molecular absorption spectrometer. The sensitivity of the classical reaction between Cr^6+^ and diphenylcarbazide is greatly enhanced by the addition of two low-cost surfactants to obtain a condensed phase in which the colored product is extracted without the use of organic solvents. The concentration effect inherent in the CPE process is very high so that the limit of determination is as low as 0.04 µg L^−1^ chromium. A drawback of this mode of operation is the need for a centrifugation step, which makes the procedure difficult to apply in a portable system. The possibility of modifying the procedure in a way that avoids the need for the centrifugation step is now being considered in our laboratory.

## Figures and Tables

**Figure 1 molecules-29-00153-f001:**
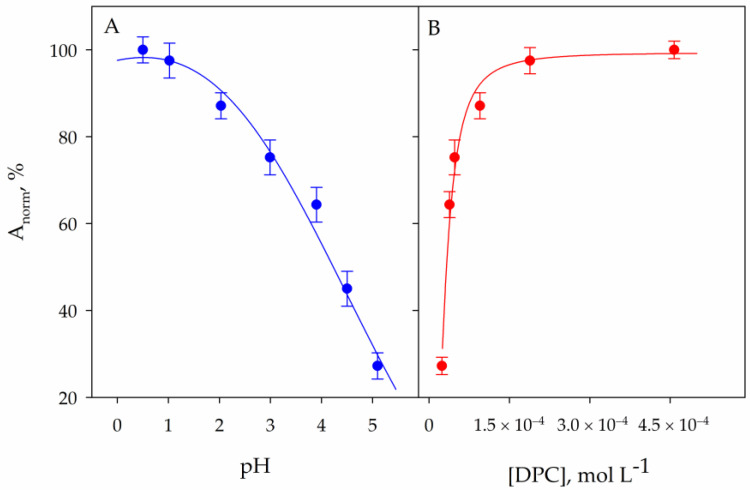
Effect of pH (**A**) and DPC concentration (**B**) on the signal obtained in aqueous solution by reaction of 1 mg L^−1^ Cr^6+^ solution with DPC. Error bars correspond to the standard deviation of three experiments.

**Figure 2 molecules-29-00153-f002:**
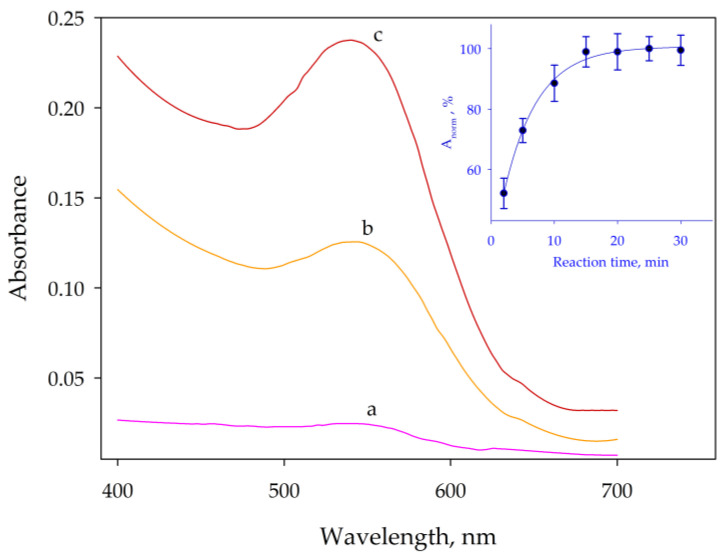
Absorption spectra of the colored complex obtained by treating 35, 150, and 300 µg L^−1^ (curves a–c, respectively) of chromium with DPC. The inset shows the kinetics of the complex formation. Error bars correspond to the standard deviation of three experiments.

**Figure 3 molecules-29-00153-f003:**
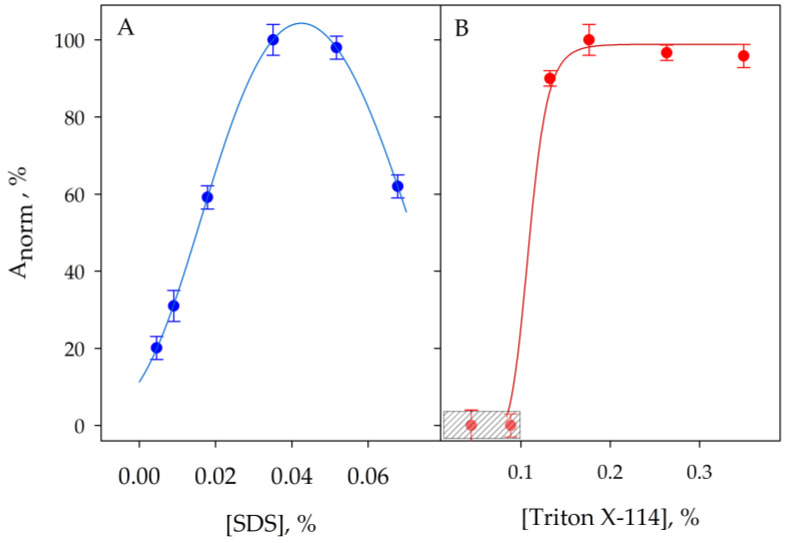
Effect of SDS (**A**) and Triton X-114 (**B**) concentrations on the signal obtained by application of the proposed procedure. The shaded area corresponds to concentrations of the surfactant where the condensed phase is not formed. Error bars correspond to the standard deviation of three experiments.

**Figure 4 molecules-29-00153-f004:**
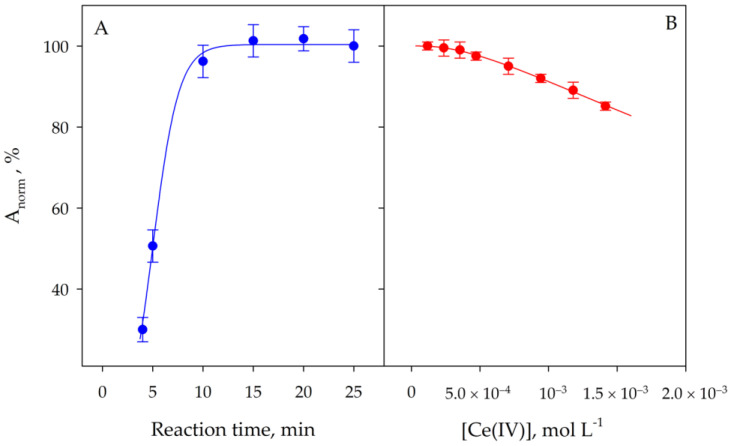
Effect of contact time (**A**) and Ce^4+^ concentration (**B**) on the signal obtained by application of the proposed procedure to 1 µg L^−1^ Cr^3+^ solutions. Error bars correspond to the standard deviation of three experiments.

**Figure 5 molecules-29-00153-f005:**
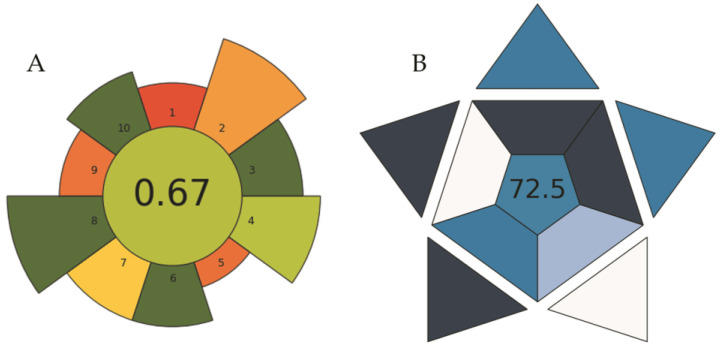
AGREEprep assessment (**A**) and BAGI index (**B**) pictograms of the proposed procedure.

**Table 2 molecules-29-00153-t002:** Study of Cr^6+^ and total chromium recovery at different Cr^6+^/Cr^3+^ concentration ratios.

Cr^6+^/Cr^3+^	Cr^6+^			Cr (Total)	
	Added, µg L^−1^	Found ^a^, µg L^−1^	Recovery, %	Found ^b^, µg L^−1^	Recovery, %
0 (0/2)	0	<LOD	–	2.01 ± 0.01	100.5
0.1 (0.2/2)	0.2	0.19 ± 0.01	97.5	2.19 ± 0.02	99.5
1 (0.5/0.5)	0.5	0.50 ± 0.01	100.4	1.02 ± 0.01	102.0
5 (1/0.2)	1	1.02 ± 0.01	102.0	1.21 ± 0.01	100.8
10 (2/0.2)	2	1.99 ± 0.02	99.5	2.19 ± 0.02	99.5

^a^ using the proposed procedure (*n* = 3); ^b^ prior oxidation with Ce^4+^.

**Table 3 molecules-29-00153-t003:** Analytical results obtained in the determination of Cr^3+^ and Cr^6+^ in water samples.

Sample	Added ^a^, µg L^−1^		Found ^a^, µg L^−1^			Recovery, %	
	Cr^3+^	Cr^6+^	Cr^3+^	Cr^6+^	Cr (Total)	Cr^3+^	Cr^6+^
Tap water	0	0	<LOD	<LOD	<LOD	-	-
	0.1	0.1	0.10 ± 0.01	0.09 ± 0.01	0.19 ± 0.01	101	94
	0.5	0.5	0.49 ± 0.01	0.49 ± 0.02	0.98 ± 0.01	99	96
Spring water	0	0	<LOD	<LOD	<LOD	-	-
	0.1	0.1	0.11 ± 0.01	0.09 ± 0.01	0.20 ± 0.01	105	97
	0.5	0.5	0.48 ± 0.02	0.49 ± 0.02	0.97 ± 0.02	92	98
Bottled water 1	0	0	<LOD	<LOD	<LOD	-	-
	0.1	0.1	0.09 ± 0.02	0.11 ± 0.01	0.20 ± 0.02	96	106
	0.5	0.5	0.51 ± 0.02	0.49 ± 0.02	1.00 ± 0.02	105	95
Bottled water 2	0	0	0.05 ± 0.01	0.03 ± 0.01	0.08 ± 0.01	-	-
	0.1	0.1	0.14 ± 0.02	0.13 ± 0.02	0.27 ± 0.02	93	89
	0.5	0.5	0.56 ± 0.03	0.51 ± 0.03	1.07 ± 0.03	102	96

^a^ Mean value of three determinations ± standard deviation.

**Table 4 molecules-29-00153-t004:** Analytical results obtained in the determination of Cr^3+^ and Cr^6+^ in reference materials.

SRM	Dilution	Certified	Cr Found ^a^, µg L^−1^		
		Total, µg L^−1^	Cr^3+^	Cr^6+^	Cr (Total)
1640 ^a^	1:100	40.22 ± 0.28	15.2 ± 0.1	27.1 ± 0.2	42.3 ± 0.1
TM-23.4	1:5	6.77 ± 0.63	6.05 ± 0.03	0.05 ± 0.01	6.10 ± 0.09
TM-25.4	1:20	24.0 ± 1.73	23.6 ± 0.1	0.07 ± 0.01	24.3 ± 0.1
TMRain-04	1:1	0.866 ± 0.165	0.85 ± 0.05	<LOD	0.85 ± 0.05

^a^ Mean value of three determinations ± standard deviation.

## Data Availability

The data are contained within the manuscript.

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
