# Peer review of "Improvement in the Chromium(VI)-Diphenylcarbazide Determination Using Cloud Point Microextraction; Speciation of Chromium at Low Levels in Water Samples"

_molecules, 2023, doi:10.3390/molecules29010153_

Round 1

Reviewer 1 Report

Comments and Suggestions for Authors

The manuscript "Improvement of the Cr (VI)-diphenylcarbazide determination using cloud point microextraction. Speciation of chromium at low levels in water samples" describes a new preconcentration procedure designed for quantitative determination of Cr6+ species. The need in such measures is caused by the fact that the standard procedure of Cr2O72- or CrO42- ions with diphenylcarbazide is selective but not very sensitive. The procedure proposed includes accumulation of all chromium species in coacervate that can be separated from the solution and posess high analyte concentration making the consequtive determination much easier. In addition, an oxidative stage with Ce4+ can be introduced to transform Cr3+ ions into hexavalent species, which allows for determining the total chromium content. The disadvantages of the method designed are the need in centrifuging (which limits the applicability of method by labs) and relatively long time of analysis, which is caused by slow kinetics of diphenylcarbazide oxidation. Therefore, the search for some catalyst for this analytic reaction to make the method more express can also be recommended.

Minor points:

1. Why SDS at concentrations higher than 0.05% leads to the decrease in the completeness of the analyte reaction (Fig. 3A)?

2. Why Ce4+ at higher concentrations also leads to the decrease in the analyte reaction yield?

3. According to the IUPAC recommendations, ions should be denoted as either Cr3+ or chromium(III), but not Cr(III).

Author Response

Dear Reviewer:

According to your comments, we have carried out a revision of our manuscript (molecules-2782487: Improvement of the chromium(VI)-diphenylcarbazide determination using cloud point microextraction. Speciation of chromium at low levels in water samples), carefully addressing all the comments. The original text is reproduced in bold, and the answer comes immediately after each point. The authors want to thank the reviewer for the suggestions, comments, and corrections, which definitely helped to improve the quality of the final version of the manuscript.

Reviewer 1 comments:

…..The disadvantages of the method designed are the need in centrifuging (which limits the applicability of method by labs) and relatively long time of analysis, which is caused by slow kinetics of diphenylcarbazide oxidation. Therefore, the search for some catalyst for this analytic reaction to make the method more express can also be recommended.

The disadvantage of the centrifugation step in the proposed procedure has already been pointed out by the authors in the Conclusions section. We are working on the development of a new procedure that does not require the use of centrifugation for phase separation. On the other hand, the observation on the use of catalysts in the kinetics of the oxidation reaction of diphenylcarbazide by chromium(VI) has been included on page 3, lines 134-136.

  1. Why SDS at concentrations higher than 0.05% leads to the decrease in the completeness of the analyte reaction (Fig. 3A)?

The response to this comment is included on page 4, lines 169-170.

  1. Why Ce4+ at higher concentrations also leads to the decrease in the analyte reaction yield?

The response to this comment is included on page 6, lines 260-2261.

  1. According to the IUPAC recommendations, ions should be denoted as either Cr3+ or chromium(III), but not Cr(III).

Your suggestion was taken into account and the document was completely corrected.

Reviewer 2 Report

Comments and Suggestions for Authors

The manuscript is on the improvement of the Cr (VI)-diphenylcarbazide determination using cloud point microextraction. Speciation of chromium at low levels in water samples.

It is well written and organised however it can be improved by taking into account the following comments:

Table 3 should be given in one page rather than splitting in two pages.

The greeness of the method should be evaluated by using a metric tool eg Agreeprep or complexGAPI

The practicality/applicability assessent of the method compared to already existing in literature could support the conclusions. This can be easily made by BAGI tool.

Author Response

Dear Reviewer:

According to your comments, we have carried out a revision of our manuscript (molecules-2782487: Improvement of the chromium(VI)-diphenylcarbazide determination using cloud point microextraction. Speciation of chromium at low levels in water samples), carefully addressing all the comments. The original text is reproduced in bold, and the answer comes immediately after each point. The authors want to thank the reviewer for the suggestions, comments, and corrections, which definitely helped to improve the quality of the final version of the manuscript.

Reviewer 2 comments:

  1. Table 3 should be given in one page rather than splitting in two pages.

In the original document, Table 3 was presented in its entirety on one page. We have rearranged the text so that this is the case in the revised version.

  1. The greeness of the method should be evaluated by using a metric tool eg Agreeprep or complexGAPI

The response to this comment is included on page 7, lines 349-353. A new Figure 5A has been included.

  1. The practicality/applicability assessent of the method compared to already existing in literature could support the conclusions. This can be easily made by BAGI tool.

The response to this comment is included on page 7, lines 354-361. A new Figure 5B has been included.
